# Vivid-VR: Distilling Concepts from Text-to-Video Diffusion Transformer for Photorealistic Video Restoration

**Haoran Bai, Xiaoxu Chen, Canqian Yang, Zongyao He, Sibin Deng & Ying Chen**[*]
Alibaba Group, China

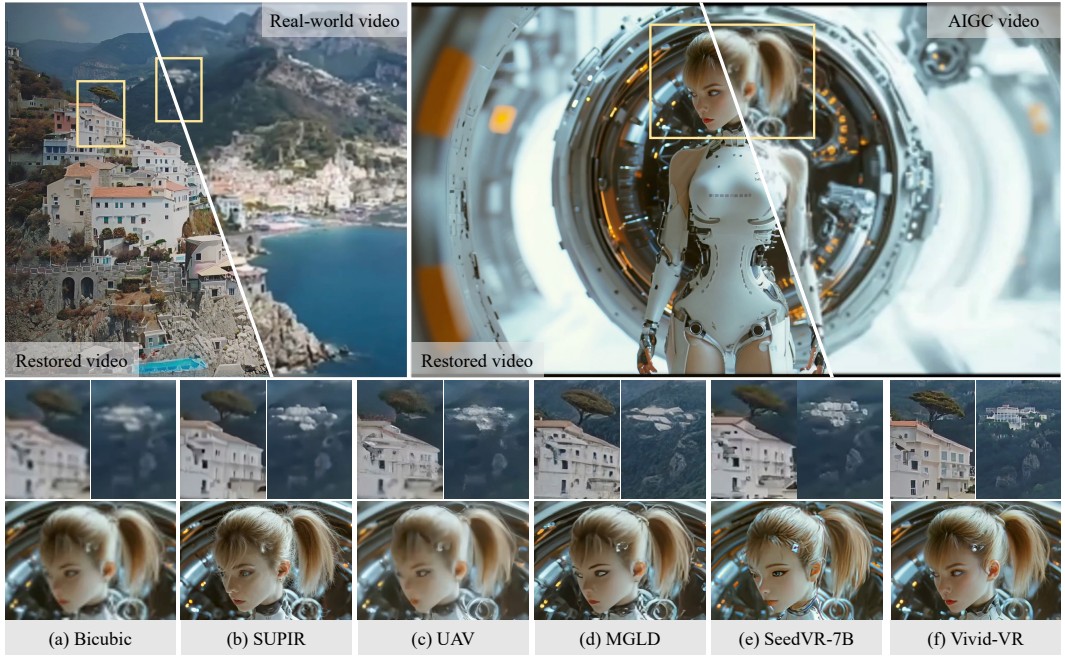

| (a) Bicubic | (b) SUPIR | (c) UAV | (d) MGLD | (e) SeedVR-7B | (f) Vivid-VR |

Figure 1: Video restoration results on both real-world and AIGC videos. To mitigate drift during fine-tuning of the controllable generation pipeline, we propose a concept distillation strategy that preserves both texture realism and temporal coherence in generated videos. Leveraging this strategy, Vivid-VR achieves impressive texture realism and visual vividness. (**Zoom-in for best view**)

## Abstract

We present Vivid-VR, a DiT-based generative video restoration method built upon an advanced T2V foundation model, where ControlNet is leveraged to control the generation process, ensuring content consistency. However, conventional fine-tuning of such controllable pipelines frequently suffers from distribution drift due to limitations in imperfect multimodal alignment, resulting in compromised texture realism and temporal coherence. To tackle this challenge, we propose a concept distillation training strategy that utilizes the pretrained T2V model to synthesize training samples with embedded textual concepts, thereby distilling its conceptual understanding to preserve texture and temporal quality. To enhance generation controllability, we redesign the control architecture with two key components: 1) a control feature projector that filters degradation artifacts from input video latents to minimize their propagation through the generation pipeline, and 2) a new ControlNet connector employing a dual-branch design. This connector synergistically combines MLP-based feature mapping with cross-attention mechanism for dynamic control feature retrieval, enabling both content preservation and adaptive control signal modulation. Extensive experiments show that Vivid-VR performs favorably against existing approaches on both synthetic and real-world benchmarks, as well as AIGC videos, achieving impressive texture realism, vi-

---

[*]Corresponding author

sual vividness, and temporal consistency. The codes and checkpoints are publicly available at https://github.com/csbhr/Vivid-VR.

# 1 INTRODUCTION

Video restoration aims to recover lost textures, fine details, and structural information from low-quality (LQ) input videos to produce high-quality (HQ) ones. Traditional reconstruction-based methods typically employ CNNs Wang et al. (2019); Pan et al. (2021); Chan et al. (2021; 2022a;b) and Transformers Liang et al. (2024; 2022) to extract visual cues for quality enhancement. However, these approaches face inherent limitations due to insufficient prior knowledge and the ill-posed nature of the inverse problem, reconstructing high-quality textures directly from severely degraded inputs remains extremely challenging. While GAN-based methods Wang et al. (2018; 2021) can generate some textures to a certain extent, their generative capacity remains limited.

Recent years have witnessed significant advancements in diffusion-based generative models Rombach et al. (2022); Podell et al. (2023); Blattmann et al. (2023), which can now synthesize photorealistic content. This progress has established generative video restoration as a promising new paradigm. While initial explorations using text-to-image (T2I) diffusion models have shown impressive results in image restoration tasks Wang et al. (2024); Yu et al. (2024); Chen et al. (2025a), their direct application to video sequences suffers from significant temporal inconsistencies due to inadequate motion modeling. Early attempts to address this limitation typically incorporate temporal enhancement mechanisms, including adding trainable temporal layers to diffusion denoisers and VAE decoders Zhou et al. (2024), or employing optical flow-based motion compensation Yang et al. (2024a). However, these post-modifications during model fine-tuning are insufficient for achieving robust temporal coherence. The advent of Diffusion Transformers (DiT) Peebles & Xie (2023) has enabled a significant leap forward, with text-to-video (T2V) models Yang et al. (2024b) now capable of generating both high-quality and temporally stable video content. This has spurred the development of T2V-based restoration approaches. For instance, SeedVR Wang et al. (2025b) integrates the shift-window attention mechanism with DiT for computational efficiency, and STAR Xie et al. (2025) proposes a dynamic frequency loss for enhanced fidelity, both achieving decent results.

Despite their advancements, current restoration methods still underperform native T2V models in both texture realism and temporal coherence. This performance gap stems primarily from distribution drift induced by imperfect multimodal alignment during the fine-tuning process. This issue is not prominent in the T2V model pretraining phase because of the large, diverse training dataset. But the challenge becomes significantly amplified when fine-tuning these models for video restoration, manifesting as unrealistic textures and compromised temporal consistency.

To overcome this challenge, we propose a concept distillation training strategy that leverages synthetic data generated by a pre-trained T2V model. The proposed approach begins with a source video and its corresponding text description obtained through a vision-language model (VLM). We first corrupt the source video with noise, then employ the pre-trained T2V model to perform denoising while incorporating the text description. This process yields a video that encapsulates the T2V model's semantic understanding of the textual concepts, ensuring inherent modal alignment between the generated video and text description in the T2V model's latent space. By blending these synthesized data with real training samples during fine-tuning, our method successfully transfers the T2V model's conceptual knowledge to the video restoration model, thereby mitigating the distribution drift problem while preserving both texture realism and temporal coherence.

Furthermore, we use ControlNet Zhang et al. (2023a) for generation control and introduce two key innovations. First, we develop a control feature projector, which effectively filters degradation artifacts to minimize their propagation through the generation pipeline. While FaithDiff Chen et al. (2025a) achieves similar functionality by jointly fine-tuning the VAE encoder which is expensive to train, our solution implements this feature projector as a lightweight CNN-based extension to the VAE encoder. Second, we redesign the ControlNet connector with a dual-branch architecture. Different from existing connectors Yu et al. (2024) which fail to properly consider DiT features during fusion, we combine an MLP branch with a cross-attention mechanism, enabling dynamic feature retrieval that preserves the generation quality and realism of native T2V models. Benefiting from these improvements, the proposed method, named Vivid-VR, achieves impressive texture realism and visual vividness (see Figure 1).

In summary, our main contributions are as follows:

- We propose a novel concept distillation training strategy that leverages a pre-trained T2V model to synthesize aligned text-video pairs, effectively mitigating distribution drift during fine-tuning and preserving texture and temporal quality.
- We improve the ControlNet architecture by introducing a lightweight control feature projector and a dual-branch connector, enabling degradation artifact removal and dynamic control feature retrieval.
- The proposed Vivid-VR performs favorably against existing methods on both synthetic and real-world benchmarks, as well as AIGC videos.

## 2 RELATED WORK

**Reconstruction-based Video Restoration.** Early approaches focused on architecture design and loss functions for direct HQ reconstruction from degraded inputs. CNN-based methods employed various strategies for temporal information integration, including optical flow estimation Caballero et al. (2017); Pan et al. (2020), deformable convolutions Wang et al. (2019), bidirectional feature propagation Chan et al. (2021), and optical flow-guided deformable alignment modules Chan et al. (2022a). Transformer-based methods Liang et al. (2024; 2022) improved performance through attention mechanisms for long-term spatio-temporal modeling. Meanwhile, some studies Wang et al. (2021); Zhang et al. (2021) have introduced more complex degradation simulations to improve real-world generalization. To produce richer textural details, GAN-based frameworks Wang et al. (2018; 2021) are consequently adopted that incorporate adversarial training. Despite these advances, methods relying solely on input-derived cues without strong priors still produce overly smoothed results when handling severely degraded content.

**Diffusion-based Video Restoration.** Diffusion-based generative models Rombach et al. (2022); Podell et al. (2023); Yang et al. (2024b) have made significant progress, which introduce a new paradigm for restoration tasks. Initial explorations focused on image restoration Wang et al. (2024); Yu et al. (2024); Chen et al. (2025a), and they achieved remarkable results. However, these approaches fundamentally lack temporal modeling capabilities, resulting in severe frame inconsistencies when directly applied to video sequences. Early solutions Zhou et al. (2024); Yang et al. (2024a) attempted to mitigate this through temporal enhancement techniques, such as incorporating trainable temporal layers or implementing optical flow-based motion compensation, yet these post-hoc adjustments proved inadequate for ensuring robust temporal coherence. Diffusion Transformers (DiT) Peebles & Xie (2023) enabled high-quality T2V generation Yang et al. (2024b) with superior temporal stability, inspiring video restoration methods. SeedVR Wang et al. (2025b) combines the shift-window attention mechanism with DiT to improve computational efficiency. STAR Xie et al. (2025) designs a dynamic frequency loss function to improve fidelity. Concurrently, efforts to improve inference efficiency have led to the design of one-step diffusion models Wang et al. (2025a); Chen et al. (2025b). Nevertheless, existing methods still exhibit noticeable gaps in texture realism and temporal consistency compared to native T2V models, due to distribution drift from imperfect multimodal alignment of training data. To bridge this gap, our work introduces a concept distillation training strategy that effectively preserves the texture and temproal quality of the base T2V model.

## 3 METHOD

The proposed Vivid-VR leverages the advanced T2V model (i.e., CogVideoX1.5-5B Yang et al. (2024b)) as its foundation, incorporating the ControlNet Zhang et al. (2023a) to condition the generation process on input videos. Figure 2 shows an overview of the proposed method. In this section, we first present the model architecture of the proposed method, and then explain the proposed concept distillation training strategy.

### 3.1 MODEL ARCHITECTURE

**Text Description Generation.** Building upon the T2V-based framework, the proposed method requires both LQ input video and corresponding text descriptions. We employ CogVLM2-Video Yang et al. (2024b) for text generation to maintain consistency with CogVideoX1.5-5B's training config-

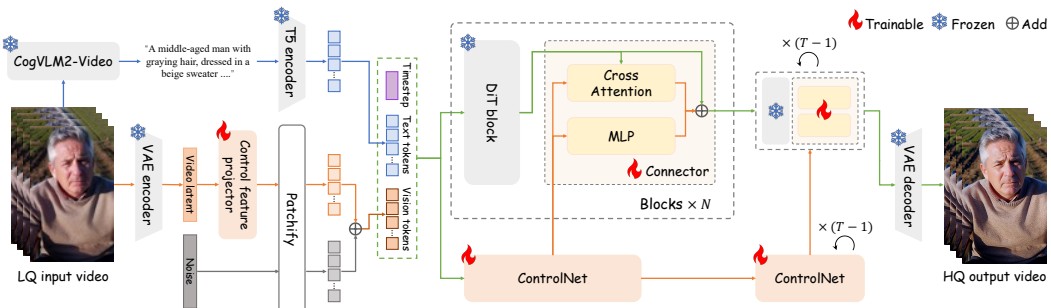

Figure 2: An overview of the proposed method. Vivid-VR first processes the LQ input video with CogVLM2-Video to generate a text description, which is encoded into text tokens via T5 encoder. Simultaneously, the 3D VAE encoder converts the input video into latent, where our control feature projector removes degradation artifacts. The video latent is then patchified, noised, and combined with text tokens and timestep embeddings as input to DiT and ControlNet. For enhanced controllability, we introduce a dual-branch connector: an MLP for feature mapping and a cross-attention branch for dynamic control feature retrieval. After $T$ denoising steps, the 3D VAE decoder reconstructs the HQ output. Only the control feature projector, ControlNet, and connectors are trained via the proposed concept distillation strategy, and other parameters remain frozen.

uration. Given the input LQ video, CogVLM2-Video produces an aligned text description, subsequently encoded into text tokens through the T5 Raffel et al. (2020) text encoder.

**Control Feature Preprocessing.** In parallel with text tokens generation, we preprocess the LQ input video to generate corresponding visual tokens for DiT and ControlNet. The preprocessing pipeline begins by encoding LQ video through the VAE encoder, producing the latent representation that contains both content information and degradation artifacts. Since these degradation artifacts may negatively impact generation quality, we propose a lightweight *Control Feature Projector* to eliminate them. The proposed projector consists of three cascaded spatiotemporal residual blocks that effectively filter the degraded features, outputting a cleaner latent representation. The video latent is then patchified and noise injected to form the visual tokens for subsequent processing.

**ControlNet Pipeline.** Given the text tokens, the visual tokens and the timestep embedding, DiT and ControlNet both perform $T$ denoising steps. DiT comprises $N$ DiT blocks, while ControlNet contains $N/7$ blocks initialized from DiT's first $N/7$ ones. During denoising process, ControlNet's visual tokens are integrated into DiT through $N$ proposed *Dual-branch Connectors*. For the $i^{th}$ connector, the fusion process of the control visual tokens is:

$$\hat{f}^i = f^i + MLP(c^{\lfloor i/7 \rfloor}) + CA(f^i, c^{\lfloor i/7 \rfloor}), \tag{1}$$

where $f^i$ denotes the visual tokens from the $i^{th}$ DiT block; $c^{\lfloor i/7 \rfloor}$ represents the corresponding ControlNet block visual tokens aligned with the $i^{th}$ DiT block; $MLP(\cdot)$ and $CA(\cdot)$ are the MLP layer and cross attention module respectively; $\hat{f}^i$ is the fused visual tokens. After $T$ denoising steps, the visual tokens are unpatchified and fed into the VAE decoder to generate the final HQ outputs.

## 3.2 Concept Distillation Training Strategy

**Training Data Collection.** Effective training of DiT-based video restoration models demands extensive high-quality text-video pairs, but existing public datasets Su et al. (2017); Nah et al. (2019); Xue et al. (2019); Stergiou & Poppe (2022) lack in both scale and diversity. To address this, we collected a large-scale video pool consisting of 3 million videos with resolutions higher than $1024 \times 1024$, frame rates higher than $24$, and durations higher than 2 seconds. These videos cover a wide range of scenes, including portraits, natural landscapes, plants and animals, urban landscapes, etc. To ensure video quality, we further screened these videos using the no-reference video quality assessment metrics Wu et al. (2023); Zhang et al. (2023b) to remove low-quality videos. For the remaining HQ videos, we generated text descriptions using CogVLM2-Video Yang et al. (2024b), maintaining consistency with CogVideoX1.5-5B's configuration. The final curated multimodal training dataset comprises $500K$ text-video pairs with exceptional quality and variety.

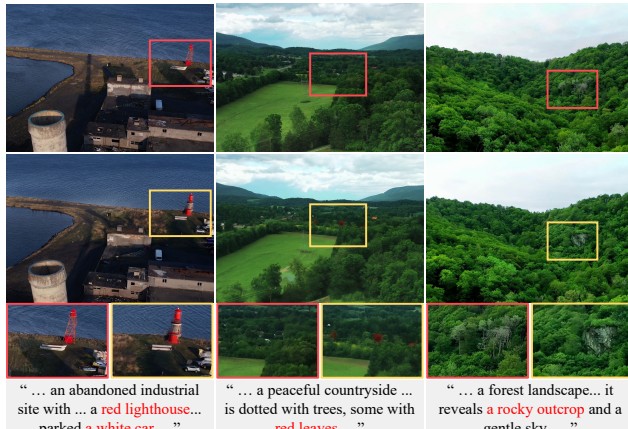

Figure 3: Example videos generated by the proposed concept distillation training strategy. The top row presents source videos, and the second row shows corresponding generated videos after embedding textual concepts via the T2V model. Due to VLM captioner limitations, the source videos exhibit imperfect alignment with their text descriptions, while the generated videos have better modality alignment. (**Zoom-in for best view**)

**Concept Distillation.** Due to limitations of the VLM captioner, the constructed text-video data pairs are not perfectly aligned (see Figure 3 top row). This may lead to distribution drift during fine-tuning, degrading output video quality. While developing a more accurate VLM captioner could enhance text-video alignment, it has two drawbacks: (1) it is costly, and (2) discrepancies may persist in the T2V model's latent space, potentially still leading to distribution drift. Instead, we address this issue by distilling the text-concept understanding capabilities of the T2V base model into the video restoration model. To this end, we employ the T2V model itself (i.e., CogVideoX1.5-5B) to perform text-guided video-to-video translation, generating training data for distillation. Specifically, given a text-video pair, we perturb the source video by adding noise with a standard deviation corresponding to $T/2$ time steps. We then apply CogVideoX1.5-5B to denoise the video over $T/2$ steps, conditioned on the text description, yielding a synthesized video with inherent alignment to text concepts in this T2V model's latent space. As illustrated in Figure 3 (second row), the generated video largely retains the source content, but some concepts have been modified to better align with those in the text description. We randomly extract text-video pairs from the constructed multimodal training dataset and employ the aforementioned process to generate $100K$ sample pairs. These generated samples are then combined with the original training dataset to facilitate fine-tuning of the control module in our DiT-based video restoration model.

**Model Training.** Following the settings of CogVideoX1.5-5B Yang et al. (2024b), we employ $v$-prediction for training, and the loss function is:

$$\mathcal{L} = \mathbb{E}_{x_0,t,\epsilon}\left[\left\|v - v_\theta(x_t, x^{lq}, x^{text}, t)\right\|^2\right], \tag{2}$$

where $x_0$ is the HQ video sampled from training dataset; $x^{text}$ and $x^{lq}$ are the corresponding text description and the synthesized LQ video using the degradation model from Wang et al. (2021); $t$ and $\epsilon$ are the time step and noise; $x_t = \sqrt{\bar{\alpha}_t}x_0 + \sqrt{1-\bar{\alpha}_t}\epsilon$ is the noised latent of $x_0$, and $\bar{\alpha}_t$ is the cumulative multiplication of the variance corresponding to time step $t$; $v_\theta$ denotes the networks of DiT and ControlNet (including the control feature projector and the connectors); $v$ is the optimization target, which is defined as $v = \sqrt{\bar{\alpha}_t}\epsilon - \sqrt{1-\bar{\alpha}_t}x_0$. During training, only the control feature projector, ControlNet, and connectors are trained, and other parameters remain frozen.

## 4 EXPERIMENTAL RESULTS

In this section, we evaluate the proposed Vivid-VR on both synthetic and real-world benchmark datasets and compare it with state-of-the-art methods.

### 4.1 IMPLEMENTATION DETAILS

The overall training dataset includes $500K$ real videos and $100K$ generated videos, as well as their corresponding text descriptions. During training, we resize the short side of these videos to $1024$ pixels and then center-crop them to $1024 \times 1024$ resolution. The number of training video frames is randomly selected between 17 and 37. We use the AdamW optimizer Loshchilov & Hutter (2017) with a learning rate of 0.0001, and adopt cosine annealing learning rate scheme Wang et al. (2019).

Table 1: Quantitative comparisons on benchmarks, including synthetic (SPMCS, UDM10, YouHQ40), real-world (VideoLQ, UGC50), and AIGC (AIGC50) videos. The best and second performances are marked in red and blue, respectively.

| Datasets | Metrics | Real-ESRGAN | SUPIR | MGLD | UAV | STAR | DOVE | SeedVR-7B | SeedVR2-7B | Vivid-VR |
|---|---|---|---|---|---|---|---|---|---|---|
| SPMCS | PSNR ↑ | 23.19 | 21.86 | 21.02 | 23.01 | 24.18 | 24.80 | 24.08 | 26.07 | 21.73 |
| | SSIM ↑ | 0.690 | 0.609 | 0.595 | 0.606 | 0.720 | 0.754 | 0.689 | 0.777 | 0.604 |
| | LPIPS ↓ | 0.230 | 0.304 | 0.281 | 0.277 | 0.301 | 0.168 | 0.263 | 0.191 | 0.278 |
| | NIQE ↓ | 5.393 | 3.494 | 3.790 | 3.503 | 7.058 | 4.031 | 4.514 | 4.969 | 3.457 |
| | MUSIQ ↑ | 51.39 | 65.23 | 58.02 | 66.11 | 30.62 | 63.29 | 56.99 | 53.23 | 70.03 |
| | CLIP-IQA ↑ | 0.306 | 0.469 | 0.357 | 0.427 | 0.254 | 0.410 | 0.347 | 0.325 | 0.483 |
| | DOVER ↑ | 8.235 | 10.07 | 7.981 | 8.987 | 4.266 | 9.898 | 9.779 | 8.625 | 11.35 |
| | MD-VQA ↑ | 79.16 | 82.88 | 78.92 | 81.90 | 74.87 | 83.07 | 79.56 | 78.78 | 86.55 |
| UDM10 | PSNR ↑ | 27.57 | 27.02 | 28.97 | 28.20 | 27.29 | 30.53 | 27.80 | 29.04 | 24.54 |
| | SSIM ↑ | 0.857 | 0.816 | 0.873 | 0.826 | 0.855 | 0.894 | 0.848 | 0.884 | 0.761 |
| | LPIPS ↓ | 0.187 | 0.208 | 0.158 | 0.196 | 0.167 | 0.101 | 0.148 | 0.117 | 0.243 |
| | NIQE ↓ | 5.835 | 4.438 | 4.827 | 5.109 | 6.072 | 5.055 | 5.345 | 5.641 | 4.046 |
| | MUSIQ ↑ | 52.32 | 60.84 | 55.82 | 56.19 | 45.38 | 55.17 | 50.29 | 48.91 | 64.71 |
| | CLIP-IQA ↑ | 0.330 | 0.418 | 0.339 | 0.333 | 0.289 | 0.340 | 0.273 | 0.272 | 0.426 |
| | DOVER ↑ | 9.402 | 10.49 | 9.319 | 9.774 | 9.454 | 10.41 | 9.349 | 8.752 | 11.97 |
| | MD-VQA ↑ | 83.51 | 85.21 | 83.89 | 83.14 | 82.10 | 83.99 | 80.15 | 79.88 | 90.05 |
| YouHQ40 | PSNR ↑ | 23.02 | 21.57 | 23.24 | 22.31 | 22.92 | 24.10 | 22.46 | 24.00 | 21.31 |
| | SSIM ↑ | 0.655 | 0.585 | 0.639 | 0.592 | 0.657 | 0.688 | 0.621 | 0.693 | 0.579 |
| | LPIPS ↓ | 0.341 | 0.347 | 0.350 | 0.340 | 0.433 | 0.283 | 0.240 | 0.185 | 0.357 |
| | NIQE ↓ | 4.316 | 3.299 | 4.038 | 3.127 | 6.744 | 4.456 | 4.243 | 4.576 | 3.410 |
| | MUSIQ ↑ | 60.03 | 68.46 | 59.40 | 65.97 | 36.36 | 60.65 | 61.91 | 59.34 | 70.55 |
| | CLIP-IQA ↑ | 0.389 | 0.485 | 0.362 | 0.427 | 0.279 | 0.356 | 0.360 | 0.336 | 0.447 |
| | DOVER ↑ | 12.60 | 12.93 | 11.01 | 12.36 | 7.868 | 12.52 | 14.00 | 12.80 | 14.61 |
| | MD-VQA ↑ | 88.85 | 89.44 | 86.24 | 87.35 | 76.89 | 86.51 | 87.51 | 85.82 | 92.92 |
| VideoLQ | NIQE ↓ | 5.014 | 4.628 | 4.565 | 4.591 | 5.789 | 5.049 | 4.994 | 5.674 | 4.371 |
| | MUSIQ ↑ | 55.29 | 54.45 | 57.70 | 55.82 | 50.52 | 55.11 | 46.49 | 43.41 | 62.47 |
| | CLIP-IQA ↑ | 0.287 | 0.299 | 0.297 | 0.262 | 0.265 | 0.271 | 0.229 | 0.220 | 0.338 |
| | DOVER ↑ | 8.453 | 8.609 | 8.830 | 7.777 | 8.758 | 8.780 | 7.240 | 6.331 | 9.743 |
| | MD-VQA ↑ | 80.50 | 77.32 | 80.67 | 78.02 | 78.56 | 79.33 | 74.80 | 73.52 | 83.14 |
| UGC50 | NIQE ↓ | 5.866 | 5.396 | 4.633 | 5.350 | 5.754 | 5.493 | 5.662 | 6.230 | 4.361 |
| | MUSIQ ↑ | 52.22 | 58.25 | 61.42 | 54.71 | 55.01 | 57.82 | 49.76 | 46.12 | 67.61 |
| | CLIP-IQA ↑ | 0.318 | 0.382 | 0.396 | 0.353 | 0.353 | 0.353 | 0.305 | 0.276 | 0.450 |
| | DOVER ↑ | 10.25 | 12.01 | 11.78 | 10.44 | 10.92 | 11.84 | 10.47 | 8.209 | 14.46 |
| | MD-VQA ↑ | 80.85 | 82.27 | 84.81 | 81.12 | 81.93 | 82.30 | 78.69 | 75.49 | 88.89 |
| AIGC50 | NIQE ↓ | 5.680 | 5.206 | 4.953 | 5.579 | 5.737 | 5.278 | 5.029 | 5.973 | 4.184 |
| | MUSIQ ↑ | 54.26 | 58.11 | 61.39 | 57.62 | 51.66 | 62.07 | 61.61 | 49.35 | 67.18 |
| | CLIP-IQA ↑ | 0.349 | 0.380 | 0.391 | 0.376 | 0.309 | 0.379 | 0.378 | 0.290 | 0.445 |
| | DOVER ↑ | 12.36 | 13.33 | 12.70 | 12.28 | 12.10 | 14.49 | 14.46 | 11.34 | 14.51 |
| | MD-VQA ↑ | 84.56 | 84.80 | 85.45 | 83.06 | 86.97 | 85.54 | 81.47 | 80.37 | 89.69 |

We train Vivid-VR on 32 NVIDIA H20-96G GPUs, with a batch size of 1 per GPU. The number of training iterations is $30K$, and the entire training process takes approximately $6K$ GPU hours. For inference, we set the number of denoising steps to 50 and used the DPM solver Lu et al. (2022). To maintain consistency with training settings, we run inference on videos at $1024 \times 1024$ resolution. For higher resolution inputs, we employ aggregation sampling Wang et al. (2024) with direct block concatenation rather than Gaussian-weighted averaging to prevent overlapping region artifacts.

## 4.2 QUANTITATIVE RESULTS

To evaluate the performance of the proposed algorithm, we compare Vivid-VR against state-of-the-art approaches, including reconstruction-based methods (Real-ESRGAN Wang et al. (2021)), generative image restoration methods (SUPIR Yu et al. (2024)), and generative video restoration methods (UAV Zhou et al. (2024), MGLD Yang et al. (2024a), STAR Xie et al. (2025), DOVE Chen et al. (2025b), SeedVR-7B Wang et al. (2025b), SeedVR2-7B Wang et al. (2025a)). The evaluation covers synthetic (SPMCS Tao et al. (2017), UDM10 Yi et al. (2019), YouHQ40 Zhou et al. (2024)) and real-world (VideoLQ Chan et al. (2022b)) benchmarks. Furthermore, we construct two test-sets, containing real-world UGC videos (UGC50) and AIGC videos (AIGC50). For real-world and AIGC videos lacking of ground truth, we employed no-reference image (NIQE Mittal et al. (2012), MUSIQ Ke et al. (2021), CLIP-IQA Wang et al. (2023)) and video quality assessments (DOVER Wu et al. (2023), MD-VQA Zhang et al. (2023b)). For synthetic benchmarks, we supplemented these metrics with full-reference evaluations (PSNR, SSIM, and LPIPS Zhang et al. (2018)).

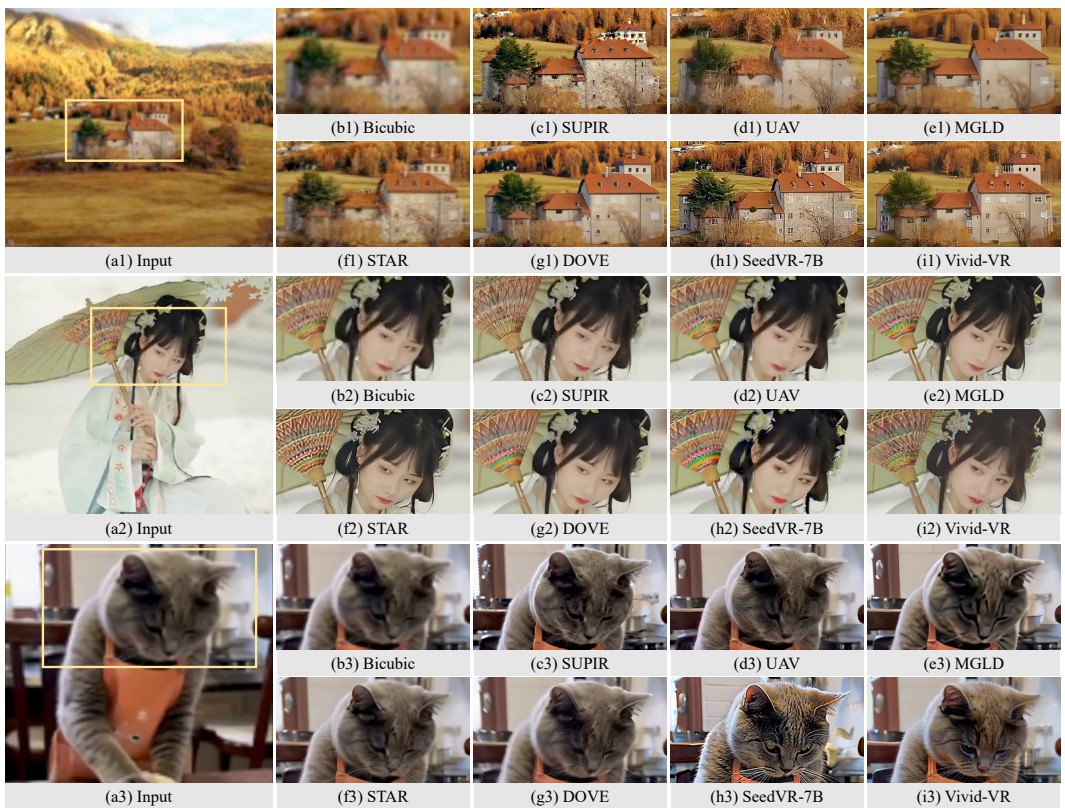

Figure 4: Qualitative comparison results on synthetic (first row), real-world (second row), and AIGC (third row) videos. The proposed Vivid-VR produces the frames with more reasonable structures, as well as more realistic and vivid textures. (**Zoom-in for best view**)

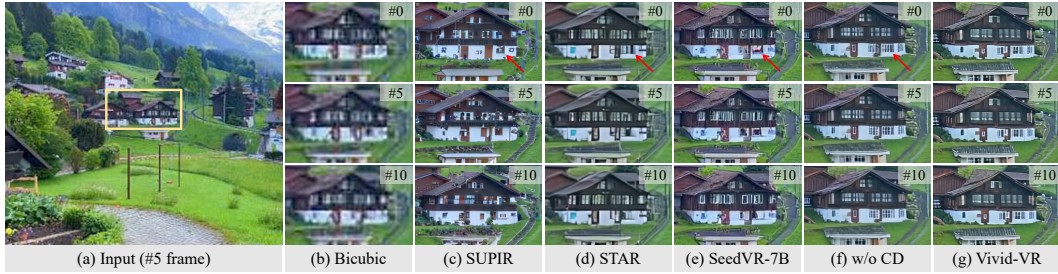

Figure 5: Visual comparison results on temporal consistency. (a) displays the #5 frame of the input video, and (b)-(g) present the outputs of the compared methods at frames 0, 5, and 10, where "CD" denotes the proposed concept distillation. Vivid-VR demonstrates superior temporal coherence, as evident from the consistent structure of windows and doors throughout the sequence. (**Zoom in on the red arrow area in each frame**)

Table 1 presents quantitative comparisons on 6 benchmark testsets. The proposed Vivid-VR significantly outperforms existing methods in no-reference metrics, achieving the best results in almost all metrics. At the same time, we also note its advantages in full-reference metrics appear less pronounced. We argue that this arises primarily from the inherent limitations of these metrics, which often fail to align with human perceptual preferences. For example, the LPIPS values of Figure 4(g1) and (i1) are 0.3112 and 0.4297, respectively, while Figure 4(i1) is more prefered by human. This phenomenon becomes particularly evident in generative restoration scenarios where severe input degradation allows multiple plausible HQ outputs, making full-reference metrics inadequate for quality assessment. This has also been mentioned in Yu et al. (2024) and has been noticed by quality assessment studies Blau & Michaeli (2018); Jinjin et al. (2020); Gu et al. (2022).

Table 2: Ablation studies of the proposed method on the UGC50 testset, where "*FT*" denotes "fine-tuning", "*CA*" denotes "cross attention", "*SK*" denotes "replacing MLP with skip connection", "*QW*" denotes "using Qwen2.5-VL as VLM captioner", "*From scratch*" denotes "synthesizing videos from scratch in concept distillation", and (i) is the setting of the proposed Vivid-VR.

| | Control Feature Preprocessing | | ControlNet Connectors | | | Concept Distillation | NIQE ↓ | MUSIQ | CLIP-IQA | DOVER |
|---|---|---|---|---|---|---|---|---|---|---|
| | *FT* VAE Enc | Projector | ZeroSFT | MLP | *CA* | | | | | |
| (a) | ✗ | ✗ | ✗ | ✓ | ✓ | ✓ | 4.622 | 63.06 | 0.414 | 13.98 |
| (b) | ✓ | ✗ | ✗ | ✓ | ✓ | ✓ | 4.632 | 64.31 | 0.408 | 14.40 |
| (c) | ✗ | ✓ | ✗ | ✓ | ✗ | ✓ | 5.183 | 59.78 | 0.374 | 13.04 |
| (d) | ✗ | ✓ | ✗ | *SK* | ✓ | ✓ | 4.730 | 63.91 | 0.401 | 13.71 |
| (e) | ✗ | ✓ | ✓ | ✗ | ✗ | ✓ | 4.771 | 61.21 | 0.389 | 13.77 |
| (f) | ✗ | ✓ | ✗ | ✓ | ✓ | ✗ | 5.364 | 57.36 | 0.363 | 12.99 |
| (g) | ✗ | ✓ | ✗ | ✓ | ✓ | *QW* | 5.253 | 60.88 | 0.354 | 13.45 |
| (h) | ✗ | ✓ | ✗ | ✓ | ✓ | *From scratch* | 4.710 | 62.66 | 0.391 | 13.27 |
| (i) | ✗ | ✓ | ✗ | ✓ | ✓ | ✓ | **4.361** | **67.61** | **0.450** | **14.46** |

## 4.3 QUALITATIVE RESULTS

Figures 1 and 4 present visual comparisons with existing methods on synthetic, real-world, and AIGC videos. The proposed Vivid-VR achieves remarkable texture realism and visual vividness. Notably, Vivid-VR is able to generate reasonable and clear structures, such as the house shown in Figure 4(i1), while existing methods exhibit structural distortions, artifacts, and loss of fine details (see Figure 4(c1)-(h1)). Moreover, the proposed Vivid-VR produces more realistic and delicate textures on human portraits and animal fur (see Figure 4(i2) and (i3)), while existing methods frequently yield either overly blurred or oversharpened outputs that are perceptually unrealistic.

In addition, Figure 5 shows visual comparisons on temporal consistency. As shown in Figure 5(g), the proposed Vivid-VR demonstrates superior coherence. For example, the structures of the windows and doors are well-consistent throughout the sequence. In contrast, SUPIR shows frame-wise inconsistency as it is an image-based restoration approach (see Figure 5(c)). While STAR and SeedVR-7B leverage T2V frameworks, their fine-tuning-induced distribution drift compromises temporal consistency (see Figure 5(d)-(e)). Notably, Vivid-VR exhibits similar degradation when without using the proposed concept distillation strategy (see Figure 5(f)).

## 5 ANALYSIS AND DISCUSSIONS

We have shown that the proposed Vivid-VR performs favorably against state-of-the-art methods. To better understand the proposed algorithm, we perform further analysis on the key components.

### 5.1 EFFECT OF THE CONTROL FEATURE PROJECTOR

The T2V base model is trained on HQ videos, making direct use of latent features from LQ videos detrimental to generation quality. To address this issue, we propose a control feature projector to remove degradation artifacts from the LQ video latents. To validate its effectiveness, we conducted an ablation study by disabling this module while keeping other settings unchanged. As shown in Table 2 ((a) vs (i)), the proposed control feature projector is able to improve video restoration. SUPIR attempts to tackle this challenge by independently fine-tuning the VAE encoder. However, this decoupled optimization creates feature incompatibility with subsequent DiT and ControlNet, leading to suboptimal results (Table 2 (b)). While this problem can be solved through joint VAE encoder and video restoration optimization, this approach is expensive to train. In contrast, our proposed lightweight control feature projector can achieve similar effects at lower training cost.

In addition, we visualize the features before and after passing through the control feature projector. As shown in Figure 6, features from the low-quality input exhibit degradation artifacts, such as blurry contours. In contrast, after being processed by our projector, these features become significantly sharper and more defined. This provides evidence that the proposed control feature projector effectively filters out degradation artifacts from the input latent signal.

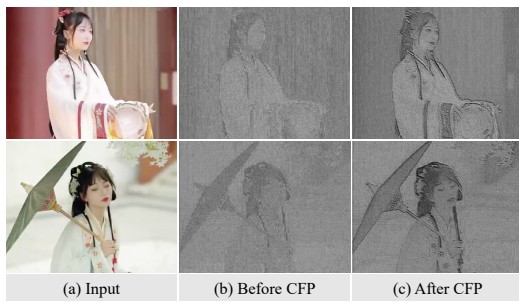

(a) Input     (b) Before CFP     (c) After CFP

Figure 6: Visualizations of features before and after passing through the control feature projector, where "CFP" denotes "control feature projector".

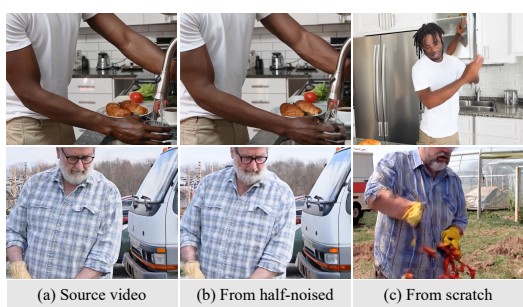

(a) Source video     (b) From half-noised     (c) From scratch

Figure 7: Example videos generated from half-noised real samples and from scratch in the proposed concept distillation process.

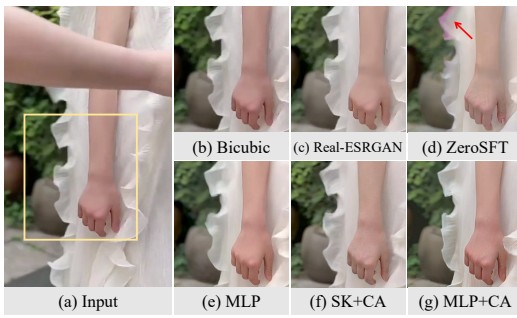

(b) Bicubic    (c) Real-ESRGAN    (d) ZeroSFT
(a) Input     (e) MLP     (f) SK+CA     (g) MLP+CA

Figure 8: Effect of the dual-branch connector, where "SK" denotes "replacing MLP with skip connection" and "CA" denotes "cross attention".

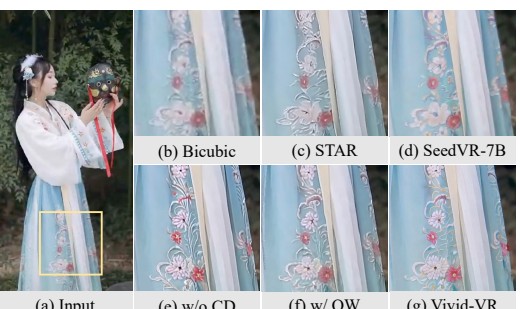

(b) Bicubic     (c) STAR     (d) SeedVR-7B
(a) Input     (e) w/o CD     (f) w/ QW     (g) Vivid-VR

Figure 9: Effect of the concept distillation, where "CD" denotes "concept distillation" and "QW" denotes "using Qwen2.5-VL as VLM captioner".

## 5.2 EFFECT OF THE DUAL-BRANCH CONNECTOR

For the ControlNet connector, we propose a dual-branch architecture combining an MLP for feature mapping with a cross attention mechanism for dynamic feature retrieval. One may wonder whether this design helps video restoration. To answer this question, we conduct three ablation studies: 1) disabling the cross attention branch; 2) replacing the MLP branch with a skip connection; 3) adopting the ZeroSFT connector Yu et al. (2024). Table 2 and Figure 8 show the comparative results of the ablation experiments. When the cross attention branch is disabled, the MLP connector does not perform well and produces results lacking in detail (see Table 2(c) and Figure 8(e)). When the MLP branch is simply disabled, the video restoration model fails to converge due to its exclusive selection of DiT-like features from control inputs, resulting in output results that do not match the input content. To ensure model convergence, we replace the MLP with a skip connection. The results in Table 2(d) and Figure 8(f) show that without the MLP feature mapping, the recovered results are not well. These experiments demonstrate the necessity of our dual-branch design.

In addition, the results in Table 2(e) show that the performance of ZeroSFT connector Yu et al. (2024) is inferior to our proposed dual-branch connector. Furthermore, the normalization operation in ZeroSFT architecture often causes residual artifacts of adjacent frames to appear in the output frames (see Figure 8(d)), while removing the normalization leads to gradient explosion during training. In contrast, our proposed connector avoids these problems.

## 5.3 EFFECT OF THE CONCEPT DISTILLATION STRATEGY

To mitigate distribution drift caused by imperfect multimodal alignment in training data, we introduce a concept distillation training strategy, that leverages the T2V base model to generate training data. To verify its effectiveness, we disable the generated training data and train this baseline method only on the collected videos. Table 2(f) and Figure 9(e) show that the baseline method without the concept distillation strategy fails to achieve high-quality results, showing overly sharp textures due to distribution drift. For the similar reason, textures generated by STAR and SeedVR are also less

Table 3: Influence of text-visual alignment in the concept distillation, where our strategy enhances text-visual alignment, and better alignment brings better restoration quality and semantic accuracy.

| | Training Data Text-Visual Alignment (FGA-BLIP2↑) | Restored Quality (DOVER↑) | Restored Semantic Accuracy (FGA-BLIP2↑) |
|---|---|---|---|
| (a) w/o Concept Distillation | 3.49 | 12.99 | 3.69 |
| (b) w/ Concept Distillation (Ours) | **3.97** | **14.46** | **3.78** |
| (c) w/ Shuffled Text during Distillation | 1.77 | 11.88 | 3.21 |

realistic. In addition, the baseline method without the concept distillation also suffers from a decline in temporal coherence (see Figure 5(f)). This verifies that our concept distillation method can facilitate video restoration in terms of both perceptual quality and temporal consistency.

Furthermore, to verify whether using a more accurate VLM captioner could resolve the distribution drift problem, we conduct an ablation study: using the more advanced Qwen2.5-VL as VLM captioner for training data annotation. As evidenced by Table 2(g) and Figure 9(f), the more accurate Qwen2.5-VL also fails to completely eliminate the modality gap in the T2V model's latent space, demonstrating the persistence of distribution drift even with superior captioning models.

In addition, one might wonder if it's possible to generate synthetic videos from scratch (starting with noise). As shown in Figure 7, we found that videos generated directly from noise often contain noticeable flaws, such as distorted human figures. Synthesizing samples from half-noised real samples significantly alleviates this issue. We further conducted an ablation study, and the results in Table 2(h) show that training a model on data generated "from scratch" degrades the video restoration performance, leading to a 1.19 point drop on the DOVER metric.

### 5.4 INFLUENCE OF TEXT-VISUAL ALIGNMENT IN CONCEPT DISTILLATION

The alignment between text and visual content is critical for concept distillation, and our method is specifically designed to enhance it. To quantitatively validate this, we employ FGA-BLIP2 Han et al. (2024) as a metric for semantic consistency between text and video. We randomly selected $10,000$ samples from the training videos generated by our concept distillation to evaluate the training data text-visual alignment. As shown in Table 3 ((a) vs (b)), the proposed concept distillation strategy boosts the training data alignment score (with the FGA-BLIP2 score increase of 0.48), which directly translates to higher quality and better semantic accuracy in the final restored videos. It is worth noting that the source videos used to generate synthetic samples are randomly sampled from our original collected dataset. This means the addition of synthetic samples does not expand or alter the underlying distribution of the training data, indicating that the performance gains stem from better text-visual alignment rather than from easier distribution of synthetic data.

Furthermore, we validate the influence of significant text-visual misalignment, where we simulated misalignment by randomly shuffling text-video pairings. As shown in Table 3(c), this drastically lowered the training data alignment (FGA-BLIP2 score dropped to 1.77). Using this data during the concept distillation process caused a significant decline in both the quality and semantic accuracy of the final output, indicating that significant text-visual misalignment degrades performance.

## 6 CONCLUSIONS AND LIMITATIONS

We have proposed Vivid-VR, a DiT-based generative video restoration method built upon an advanced T2V foundation model. To mitigate distribution drift during fine-tuning, we have introduced a concept distillation training strategy that leverages the pre-trained T2V model to synthesize training samples with embedded textual concepts. Regarding the model architecture for controllable generation, we have proposed two key components: 1) a control feature projector that removes degradation artifacts from latent video features, and 2) a dual-branch connector combining an MLP and cross-attention mechanism for control feature mapping and dynamic retrieval. Both quantitative and qualitative experimental results demonstrate the effectiveness of the proposed method.

The proposed method builds upon the CogVideoX1.5-5B T2V model and inherits its inference complexity, which results in lengthy inference times. Future work will explore ways to enhance the algorithm's efficiency, such as applying one-step diffusion fine-tuning to achieve comparable video restoration quality in a single forward pass.

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

# A APPENDIX

## A.1 USE OF LLMS

In this paper, we used LLMs to assist with grammar and writing polish. All LLM-generated suggestions were carefully reviewed and edited. The authors take full responsibility for the final content.

## A.2 ADDITIONAL ANALYSIS OF THE PROPOSED METHOD

### A.2.1 TRADE-OFF BETWEEN FIDELITY AND QUALITY

As Yu et al. (2024) points out, powerful generative prior is a double-edged sword, as excessive generative capacity may in turn affect the fidelity of the restored video. To address this, we introduce *Restoration-Guided Sampling* into Vivid-VR's inference sampling process to balance the quality and fidelity:

$$\hat{x}_t^{est} = x_t^{est} + (\frac{t}{T})^\tau (x^{lq} - x_t^{est}),\tag{3}$$

where $x^{est}$ is the denoised latent at time step $t$, and $x^{lq}$ is the original input latent; $T$ denotes the total number of denoising steps; $\tau$ is the guidance coefficient; $\hat{x}^{est}$ is the output latent after the restoration-guided sampling. Figure 10 demonstrates this trade-off: higher guidance coefficient $\tau$ yield more realistic results, while lower $\tau$ preserve greater fidelity to the original input content.

### A.2.2 EFFECT OF THE NUMBER OF GENERATED TRAINING VIDEOS

As mentioned in the main paper, the proposed method employs the concept distillation strategy to generate $100K$ videos for training. A natural question arises: does the number of generated training videos impact restoration performance? To investigate this, we conducted the ablation studies here. Table 4 demonstrates that increasing the number of generated training videos from 0 to $100K$ yields significant performance gains, while expanding from $100K$ to $150K$ shows diminishing returns. Considering the cost of generating training videos, we therefore adopt $100K$ generated videos as our standard configuration. Furthermore, we verified that relying solely on generated training data (without source videos) leads to suboptimal results. This occurs because the T2V base model's outputs contain inherent imperfections. Training exclusively on such data ultimately compromises model performance.

Table 4: Effect of the number of training videos generated by the proposed concept distillation.

| Methods | Concept Distillation Training Strategy | | NIQE ↓ | MUSIQ ↑ | CLIP-IQA ↑ | DOVER ↑ |
|---|---|---|---|---|---|---|
| | # source training videos | # generated training videos | | | | |
| (a) | 500K | 0 | 5.364 | 57.36 | 0.363 | 12.99 |
| (b) | 500K | 50K | 4.562 | 63.00 | 0.408 | 13.46 |
| (c) | 500K | 100K | 4.361 | **67.61** | **0.450** | 14.46 |
| (d) | 500K | 150K | **4.292** | 67.19 | 0.443 | **14.51** |
| (e) | 0 | 150K | 5.652 | 53.77 | 0.377 | 11.63 |

### A.2.3 EFFECT OF THE NUMBER OF CONTROLNET BLOCKS

To reduce the parameter count, we employ $N/7$ DiT blocks in ControlNet. In Deng et al. (2025), only one block is used, and all connectors share the same control feature. To further investigate whether $N/7$ DiT blocks are indeed necessary, we set the number of blocks to 1 and retrain using the same settings. The results in Table 5 show that using only one block does not perform well.

Table 5: Effect of the number of controlNet blocks.

| | NIQE ↓ | MUSIQ ↑ | CLIP-IQA ↑ | DOVER ↑ |
|---|---|---|---|---|
| 1 DiT block in ControlNet | 4.855 | 66.85 | 0.442 | 14.17 |
| $N/7$ DiT blocks in ControlNet (Vivid-VR) | **4.361** | **67.61** | **0.450** | **14.46** |

### A.2.4 EXTENDING TO OTHER PRE-TRAINED T2V MODELS

We selected CogVideoX1.5-5B as the T2V base model because it is a state-of-the-art, publicly available T2V model at the time of our research, demonstrating exceptional generation quality and temporal stability. To assess the generalizability of our approach, we replaced CogVideoX1.5-5B with a more recent open-source model, Wan2.2-TI2V-5B Wan et al. (2025). By applying the same training strategy, we found that our proposed concept distillation and ControlNet improvements transferred effectively to this new model, also achieving SOTA results (see Table 6). We also experimented with a smaller model, Wan2.1-T2V-1.3B Wan et al. (2025), and observed a corresponding drop in restoration performance. This suggests that leveraging larger and more powerful T2V models could yield superior outcomes. Extending this to even larger models (e.g., Wan2.1-T2V-14B) remains a direction for future work due to significant computational demands.

Table 6: Extending to other pre-trained T2V models.

| Pretrained T2V Model | Parameters | NIQE↓ | MUSIQ↑ | CLIP-IQA↑ | DOVER↑ |
|---|---|---|---|---|---|
| CogVideoX1.5-5B | 5B | 4.361 | 67.61 | 0.450 | **14.46** |
| Wan2.2-TI2V-5B | 5B | **4.101** | **67.88** | **0.461** | 14.01 |
| Wan2.1-T2V-1.3B | 1.3B | 5.078 | 63.21 | 0.339 | 12.98 |

### A.2.5 MORE QUALITATIVE COMPARISON

In the main paper, we have demonstrated that the proposed Vivid-VR achieves state-of-the-art performance. Due to the submission file size limit, we further provide more visual comparisons with state-of-the-art methods and video examples in the ***Supplementary Material***, where Vivid-VR demonstrates superior structural clarity, texture richness, visual vividness, and temporal consistency.

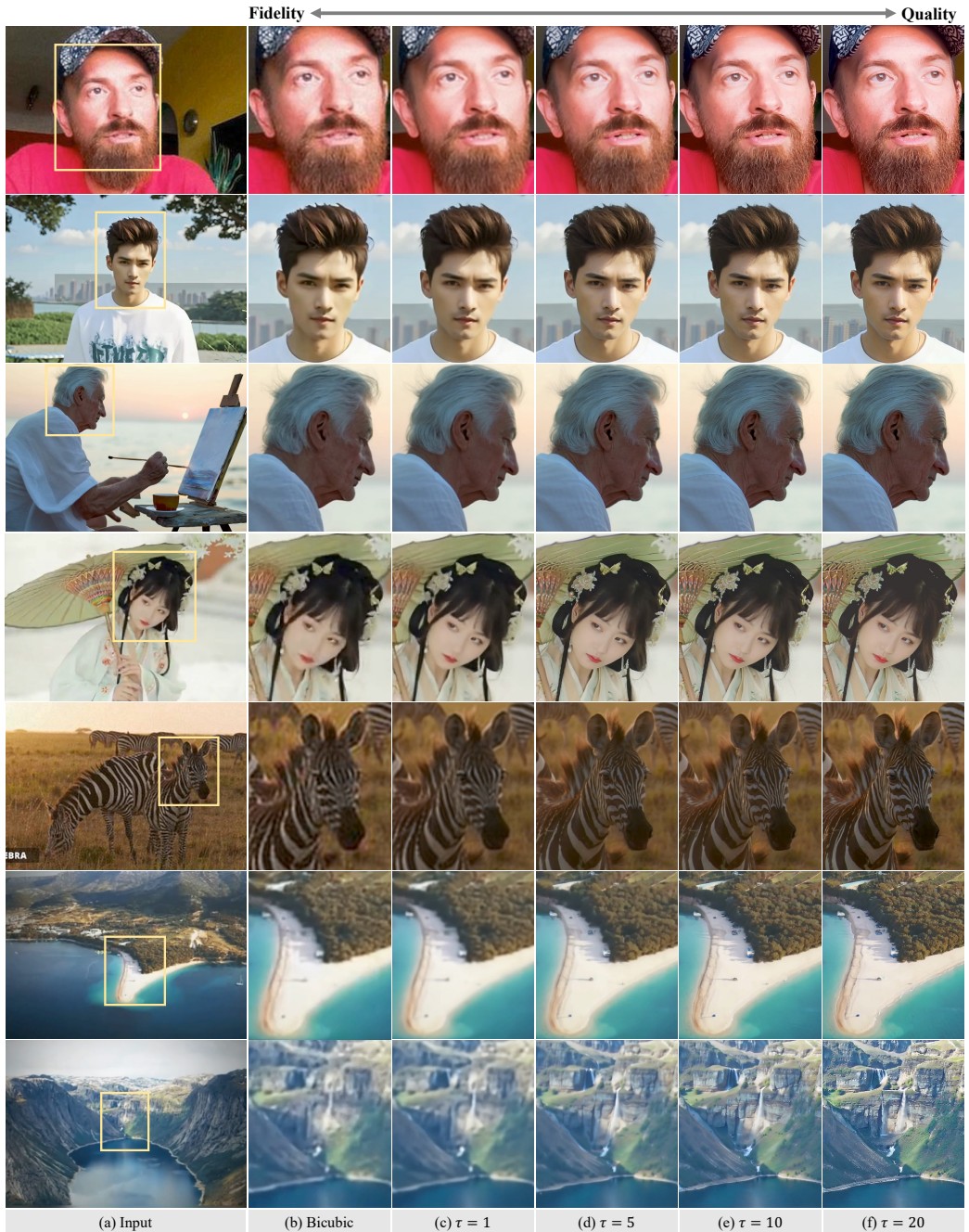

Figure 10: Trade-off between fidelity and quality. Higher guidance coefficient $\tau$ in the *Restoration-Guided Sampling* yield more realistic results, while lower $\tau$ preserve greater fidelity to the original input content. (**Zoom-in for best view**)

