# OpenReview forum: "Vivid-VR: Distilling Concepts from Text-to-Video Diffusion Transformer for Photorealistic Video Restoration"
_ICLR.cc/2026/Conference — ICLR 2026 Poster_

### Official Review · Reviewer_BT4i · 2025-10-28

**Soundness:** 3
**Presentation:** 3
**Contribution:** 2
**Rating:** 4
**Confidence:** 3

**Summary:**

This paper proposes a video restoration method. The core contributions include training with mixed real/synthesized data (termed “concept distillation”) and two architectural modifications.

**Strengths:**

The method demonstrates strong empirical performance according to the reported evaluations.

**Weaknesses:**

Technical novelty and depth are limited. The architectural modifications, while effective, are relatively standard and offer limited insight; it is unlikely that these design choices will significantly influence future research. Showing that synthesized videos can benefit video restoration is a useful observation, but this point alone, at the current level of investigation, does not meet the novelty threshold to me.

The proposed components are not sufficiently analyzed. For example, although the projector is effective, there is no evidence that its efficacy is due to “filtering degradation artifacts from input video latents” (line 49). Given the presented experiments, the safe conclusion is only that having a trainable projection layer at that location is beneficial; any stronger claims require further evidence. Regarding concept distillation, can we directly generate synthesized samples from scratch instead of starting from half-noised real samples? Moreover, how do we know that the gains from concept distillation stem from better image–text alignment rather than other factors, such as the intrinsically easier distribution of synthesized data?

**Questions:**

See weakness

---

> ### Author Response · Authors · 2025-11-20
> **Response to Weaknesses W1**
>
> Thank you for your detailed review and critical feedback. We appreciate that you recognized the strong empirical performance of our method. Your questions regarding technical novelty and the depth of our analysis are very important, and we are grateful for the opportunity to clarify our work.
>
> ## W1: Technical novelty and depth
>
> We understand your concerns regarding the technical novelty and depth of our work. However, we would like to respectfully re-frame our contribution not as a collection of incremental adjustments, but as a novel solution to a very specific and challenging problem: **how to adapt a pre-trained T2V foundation model for video restoration without suffering from distribution drift, thereby preserving its powerful generative priors**.
>
> ### >> The innovation of "Concept Distillation"
>
> You characterized this as "showing that synthesized videos can benefit video restoration." We respectfully argue the contribution is **more profound**. The core insight is that **distribution drift**, caused by a multimodal alignment gap in the T2V model's latent space, is the primary obstacle in this task. Fine-tuning on real-world data (videos captioned by a VLM) introduces this misalignment, forcing the model to deviate from its powerful priors (i.e., distribution drift) and thus degrading restoration quality.
>
> Our proposed **"Concept Distillation" is a novel strategy to address this issue** by leveraging the pre-trained T2V model itself to generate text-video pairs that are aligned well in its latent space. This approach effectively mitigates the distribution drift problem, and significantly facilitates video restoration. **This is a targeted solution to the latent space alignment problem, not merely a data augmentation trick**. We believe this framework is a key contribution that will significantly impact future work in genrative video restoration.
>
> ### >> The insight of architectural modifications
>
> While a projector or a new connector might seem like common building blocks, their design and purpose here are not arbitrary. They are motivated by solving specific problems within our framework. The Control Feature Projector offers a lightweight and practical alternative to the computationally expensive joint fine-tuning of a VAE encoder, as seen in prior work like FaithDiff. The Dual-Branch Connector is a reasoned design to explicitly balance the needs of content preservation and dynamic retrieval of control signals during controlled generation. **Their value lies not in their novelty as standalone components, but in their role as principled solutions that enable the success of our overall strategy**.

---

> ### Author Response · Authors · 2025-11-20
> **Response to Weaknesses W2**
>
> ## W2: Analysis of proposed components
>
> Thank you for these sharp and insightful questions. They push us to better justify our claims.
>
> ### >> Evidence for the claim that the projector "filter out degradation artifacts from the potential signal of the input video"
>
> To directly support this claim, we have added visualizations of the features before and after passing through the control feature projector in ***Figure 6 of our revised paper***. These visualizations show that features from the low-quality input exhibit degradation artifacts, such as blurry contours. In contrast, after being processed by our projector, these features become significantly sharper and more defined. This provides evidence that the proposed Control Feature Projector effectively filters out degradation artifacts from the input latent signal. We have add corresponding discussions in ***Section 5.1 of the revised paper***.
>
> ### >> Generate synthetic samples from scratch in Concept Distillation
>
> We found that videos generated directly from noise often contain noticeable flaws, such as distorted human figures. Synthesizing samples from half-noised real samples significantly alleviates this issue (see visualizations in ***Figure 7 of the revised paper***). We further conducted an ablation study, and the results below show that training a model on data generated "from scratch" degrades the video restoration performance, leading to a 1.19 dB drop on the DOVER metric. We have add corresponding discussions in ***Section 5.3 of the revised paper***.
>
> |  | NIQE ↓ | MUSIQ ↑ | CLIP-IQA ↑ | DOVER ↑ |
> |:-------|:-------|:-------|:-------|:-------|
> | (a) Generate from scratch |  4.710 |  62.66 | 0.391 | 13.27 |
> | (b) Generate from half-noised real samples |  **4.361** |  **67.61** | **0.450** | **14.46** |
>
> ### >> The gains from Concept Distillation stem from better video–text alignment rather than easier distribution of synthesized data
>
> We use the FGA-BLIP2 score [1] to evaluate the alignment between videos and their text descriptions.
>
> - We randomly selected 10000 samples from the training videos generated by our concept distillation to evaluate the training data text-visual alignment. The results below demonstrate that, compared to the source videos, **the videos generated via concept distillation exhibit significantly better alignment with their corresponding text** (with the FGA-BLIP2 score increase of 0.48). This provides evidence that the benefit of our strategy is attributable to enhanced video-text alignment.
> - Furthermore, as stated on *line 224 of the paper*, the source videos used to generate synthetic samples are randomly sampled from our original collected dataset. This means **the addition of synthetic samples does not expand or alter the underlying distribution of the training data**, indicating that the performance gain is not derived from easier distribution of synthetic data.
>
> We have add corresponding discussions in ***Section 5.4 of the revised paper***.
>
> |  | Text-Video Alignment (FGA-BLIP2 ↑) |
> |:-------|:-------|
> | (a) Source training videos | 3.49 |
> | (b) Generated training videos (via Concept Distillation) | **3.97** |
>
> We hope these clarifications better illuminate the novelty of our work and the depth of our analysis. We believe that by identifying and addressing the latent space alignment problem in T2V model adaptation, our work offers significant insights to the research community. Thank you again for your thoughtful feedback.
>
> ## References
>
> [1] Shuhao Han, Haotian Fan, Jiachen Fu, Liang Li, Tao Li, Junhui Cui, Yunqiu Wang, Yang Tai, Jingwei Sun, Chunle Guo, et al. Evalmuse-40k: A reliable and fine-grained benchmark with comprehensive human annotations for text-to-image generation model evaluation. arXiv preprint arXiv:2412.18150, 2024.

---

> ### Author Response · Authors · 2025-11-27
> **Following up on Paper 3680**
>
> Dear Reviewer,
>
> I hope this message finds you well.
>
> We are writing to follow up on our response to your review. As the discussion period is nearing its end with less than 7 days remaining, we wanted to ensure our rebuttal has sufficiently addressed your concerns.
>
> Your insights are invaluable to us, and we would be very grateful for any further feedback or questions you might have. We are eager to further improve our work based on your guidance.
>
> Thank you again for your time and thoughtful review.
>
> Best regards,
>
> The Authors of Paper 3680

---

> > ### Comment · Reviewer_BT4i · 2025-11-28
> > **Thanks for the rebuttal**
> >
> > Thanks to the authors for their rebuttal.
> >
> > With the added experiments, I now believe the claims are better supported. After thinking about this work for a while, I also feel that the `concept distillation' method is indeed an interesting finding (though I personally have some different understanding of why it works). Based on these points, I would like to raise my overall rating to 6 to express my recognition of the revision and rebuttal. (Note that due to system problem, I cannot update the score now)
> >
> > On the other hand, I still feel that the concept distillation method is the only qualified contribution for me (other points like architecture modifications feel more like implementation details to me). Therefore, I overall still hold a borderline attitude on whether this single point is enough to make this work worthy of acceptance. I expect the AC to consider all opinions and make a careful and balanced decision.

---

> > > ### Author Response · Authors · 2025-11-28
> > > **Thank you for raising the score on Paper 3680**
> > >
> > > We are truly grateful for your decision to raise the score. We are especially pleased to hear that our added experiments helped support our claims and, most importantly, that you found the "concept distillation" strategy to be an interesting finding. Your recognition of this core contribution means a great deal to us.
> > >
> > > We also understand and respect your perspective regarding the other components and the paper's overall contribution. Your final assessment is very helpful.
> > >
> > > Thank you again for your valuable time and guidance. It has significantly helped in strengthening our paper.
> > >
> > > Best regards,
> > >
> > > The Authors of Paper 3680

---

### Official Review · Reviewer_vgeK · 2025-10-30

**Soundness:** 3
**Presentation:** 3
**Contribution:** 3
**Rating:** 6
**Confidence:** 3

**Summary:**

Vivid-VR is a generative video restoration method. To address the issue of distribution drift during fine-tuning, the authors propose a concept distillation strategy that uses the pre-trained T2V model to synthesize aligned text-video pairs, thereby preserving texture realism and temporal coherence. The control mechanism is further enhanced with a novel feature projector to filter degradation artifacts and a dual-branch connector for dynamic control feature retrieval. Extensive experiments demonstrate that Vivid-VR achieves superior performance in texture realism and temporal consistency compared to existing methods on various benchmarks.

**Strengths:**

1. Leveraging the capabilities of pretrained T2V models to enhance the video restoration performance is an interesting approach. Using the textual description as a connector, the authors find an effective way to transfer the T2V model’s pretrained knowledge to the video restoration model, which I believe benefits the community.

2. The proposed method outperforms several previous advances by a large margin in a wide range of benchmarks. The experiments are comprehensive, and the qualitative demos are also impressive.

3. The ablation studies are also exhaustive. The effectiveness of each branch is clearly verified.

**Weaknesses:**

1. Though the idea of transferring the T2V model's capability to downstream tasks is interesting, the proposed method seems to be trivial and similar to other methods. Using a pretrained model to corrupt and reconstruct the visual content is a common way, especially in image enhancement and restoration tasks, and it is also widely adopted to add the textual description during the reconstruction. It is more likely to be a transition from the image restoration task to the video restoration task, which makes the technical contribution less competitive.

2. How the concept distillation really works remains ambiguous. Why putting the textual description into the DiT block can ''transfer the T2V model’s conceptual knowledge to the video restoration model''? What concept is transferred to the video restoration model? What will happen if we use a video captioner to generate captions and add a text encoder to the DiT block instead of leveraging the pretrained T2V model?

**Questions:**

1. For the concept distillation process, what will happen if the generated textual descriptions have a major difference from the visual content? Then the pretrained T2V model may generate videos that are quite different from the original low HQ videos. Will it influence the semantic accuracy of the final high HQ videos?

2. Why CogVideoX1.5-5B is selected as the pretrained T2V model? What will the model perform when selecting other alternatives? Besides, what will the video restoration perform when adopting pretrained T2V models in different sizes?

3. What is the rationale of selecting DiT as the main architecture of the video restoration model, considering there are many alternatives, such as MMDiT?

---

> ### Author Response · Authors · 2025-11-20
> **Response to Weaknesses**
>
> Thank you for your detailed and constructive feedback. We are encouraged that you recognized the strength of our experimental results, the comprehensive nature of our evaluations, and the value of our approach for the community. Below, we address the points raised, incorporating clarifications and new experimental evidence where relevant.
>
> ## W1: Technical contribution and perceived similarity to image restoration tasks
>
> We appreciate you raising this important point. While we agree that corrupt-and-reconstruct paradigms are established in image restoration, we respectfully argue that our work presents a non-trivial and novel solution to a problem that is unique to video restoration in the context of large-scale T2V models.
>
> - **The challenge is both spatial and temporal**. The primary failure modes when fine-tuning T2V models for video restoration include not only unrealistic textures but also the loss of **temporal consistency**, leading to flickering and incoherent object motion. This is more complex and challenging than image restoration. Our method is designed to simultaneously preserve the T2V model's strong spatial and temporal priors, which image restoration techniques do not address.
>
> - **The core problem to be solved is distribution drift, which is critical but has not yet been studied in existing methods**. This is a multimodal alignment problem between the text description and the video content within the T2V model's latent space. Fine-tuning on text-video pairs that do not perfectly match in the latent space of the T2V model forces the model to "forget" its powerful generative priors, leading to degradation in both texture and temporal coherence. Our proposed concept distillation strategy is tailored for this distribution drift problem, which we believe is a significant contribution beyond a simple transition from image task to video task.
>
> ## W2: Ambiguity of concept distillation
>
> Thank you for asking for this clarification. Let us explain the mechanism in more detail.
>
> - **Motivation**: This work investigates how to leverage the powerful generative capabilities of the pre-trained T2V model for video restoration. A common issue when fine-tuning these models is that their generative priors are often "forgotten", leading to degradation in both texture and temporal coherence. We identify that this is primarily due to the **distribution drift** problem caused by the misalignment of training text-video pairs in the T2V model's latent space. For instance, VLM can be used to generate a text description (e.g., "a cat running in a field") from the collected video, but the T2V model has its own learned prior for what a "cat," "running," and "field" should look like and how they should move together. When we fine-tune the model to restore a real video of a cat, this video may not perfectly match the T2V model's internal "concept" of the scene described by the text. This conflict is what leads to distribution drift.
>
> - **How concept distillation works**: Our strategy resolves this by synthesizing training text-video pairs using the T2V model itself. Given a source video and its text caption, we generate a target video from the text by using the T2V model (starting from a noised version of the source video). Since the generated target video is directly derived from the text caption, **such text-video pair is aligned well within the T2V model's latent space**, which encapsulates the T2V model's semantic understanding of the textual concepts. Fine-tuning the model using such pairs allows the conceptual understanding capabilities of T2V model to be transferred to the restoration model, thereby mitigating the distribution drift problem.
>
> - **What concept is transferred?** As mentioned above, the generated training text-video pairs encapsulate the T2V model's semantic understanding of textual concepts. The proposed concept distillation strategy utilizes such pairs to transfer this "**semantic understanding of textual concepts**" from T2V model to the restoration model. This is what allows our method to maintain the T2V model's powerful generative capabilities and achieving impressive texture realism and visual vividness (as shown in *Figures 1, 4, 5 of the paper*).
>
> - **What will happen if use "text encoder + DiT block" instead of "pre-trained T2V model"**? The performance of our proposed method is fundamentally based on the pre-trained T2V model, without which the text caption is ineffective. To validate this, we trained a model with the same network architecture (text encoder + DiT blocks) from scratch for video restoration, without loading the pre-trained T2V weights. When evaluated on the UGC50 testset, this model performed poorly, with its **DOVER score dropping by 3.47 points** (from 14.46 to 10.99).

---

> ### Author Response · Authors · 2025-11-20
> **Response to Questions**
>
> ## Q1: The influence of significant text-visual misalignment
>
> This is a very practical concern. The alignment between text and visual content is critical for concept distillation, and our method is specifically designed to enhance it. To quantitatively validate this, we employ FGA-BLIP2 [1] as a metric for semantic consistency between text and video. We randomly selected 10000 samples from the training videos generated by our concept distillation to evaluate the training data text-visual alignment. We conduct a series of ablation studies below, and the results demonstrate two key findings:
>
> - **Concept distillation enhances text-visual alignment**: Comparing (a) and (b), our strategy boosts the training data alignment score by 0.48, which directly translates to higher quality and better semantic accuracy in the final HQ restored videos.
>
> - **Significant text-visual misalignment degrades performance**: In (c), we simulated misalignment by randomly shuffling text-video pairings. This drastically lowered the training data alignment (FGA-BLIP2 score dropped to 1.77). Using this data during the concept distillation process caused a significant decline in both the quality and semantic accuracy of the final HQ output, indicating that significant text-visual misalignment degrades performance.
>
> We have add corresponding discussions in ***Section 5.4 of the revised paper***.
>
> |  | Training Data Alignment (FGA-BLIP2 ↑) | Restored Quality (DOVER ↑) | Restored Semantic Accuracy (FGA-BLIP2 ↑) |
> |:-------|:-------|:-------|:-------|
> | (a) w/o Concept Distillation  |  3.49  |  12.99 |  3.69 |
> | (b) w/ Concept Distillation (Ours) |  3.97  |  14.46 |  3.78 |
> | (c) w/ Shuffled Text during Distillation |  1.77  |  11.88 |  3.21 |
>
>
> ## Q2: the choice of CogVideoX1.5-5B and other alternatives
>
> We selected CogVideoX1.5-5B because it was a state-of-the-art, publicly available T2V model at the time of our research, demonstrating exceptional generation quality and temporal stability.
>
> - **Performance with alternative T2V models**: To assess the generalizability of our approach, we replaced CogVideoX1.5-5B with a recent open-source model, **Wan2.2-TI2V-5B**. By applying the same training strategy, we found that our proposed concept distillation and ControlNet improvements transferred effectively to this new model, also achieving SOTA results.
>
> - **Performance with different model scale**: We also experimented with a smaller model, **Wan2.1-T2V-1.3B**, and observed a corresponding drop in restoration performance. This suggests that leveraging larger and more powerful T2V models could yield superior outcomes. Extending this to even larger models (e.g., Wan2.1-T2V-14B) remains a direction for future work due to significant computational demands.
>
> We have add corresponding discussions in ***Appendix A.2.4 of the revised paper***.
>
> | Pretrained T2V Model | Parameters | NIQE ↓ | MUSIQ ↑ | CLIP-IQA ↑ | DOVER ↑ |
> |:-------|:-------|:-------|:-------|:-------|:-------|
> | CogVideoX1.5-5B  |  5B  |  4.361 |  67.61 | 0.450 | **14.46** |
> | Wan2.2-TI2V-5B |  5B  |  **4.101** |  **67.88** | **0.461** | 14.01 |
> | Wan2.1-T2V-1.3B |  1.3B  |  5.078 |  63.21 | 0.339 | 12.98 |
>
> ## Q3: the rationale for choosing DiT over alternatives like MM-DiT
>
> The choice of network architecture is determined by the selected pre-trained T2V model. As demonstrated in above Q2 experiments, our method generalizes well across various T2V models, each employing a different variant of the DiT architecture:
> - CogVideoX1.5-5B utilizes an Expert Transformer module, which is analogous to MM-DiT by concatenating text and visual tokens for the attention mechanism.
> - Wan2.2-TI2V-5B integrates text features into its DiT module via cross-attention.
>
> Our method performs well across these architectural variants, demonstrating its robustness and adaptability.
>
> We hope these clarifications could address your concerns. We are grateful for your feedback, which has helped us to better articulate our contributions. We would be happy to provide any further clarification.
>
> ## References
> [1] Shuhao Han, Haotian Fan, Jiachen Fu, Liang Li, Tao Li, Junhui Cui, Yunqiu Wang, Yang Tai, Jingwei Sun, Chunle Guo, et al. Evalmuse-40k: A reliable and fine-grained benchmark with comprehensive human annotations for text-to-image generation model evaluation. arXiv preprint arXiv:2412.18150, 2024.

---

> ### Author Response · Authors · 2025-11-27
> **Following up on Paper 3680**
>
> Dear Reviewer,
>
> I hope this message finds you well.
>
> We wanted to follow up and thank you again for your constructive feedback on our paper. We found your comments particularly helpful and have incorporated them into our paper and rebuttal.
>
> With the discussion period drawing to a close, we would be delighted to answer any further questions or hear any additional suggestions you might have. Our goal is to ensure the final version of the paper is as strong as possible, and your perspective is crucial to that.
>
> We truly appreciate your time and effort in helping us improve our work.
>
> Best regards,
>
> The Authors of Paper 3680

---

### Official Review · Reviewer_7MWW · 2025-10-31

**Soundness:** 3
**Presentation:** 3
**Contribution:** 3
**Rating:** 6
**Confidence:** 3

**Summary:**

The paper presents a DiT-based video restoration model trained with a novel concept-distillation strategy that uses a pre-trained T2V generator to produce aligned text–video pairs, which eliminates distribution drift. It also proposes a lightweight ControlNet projector and dual-branch connector that further suppress artifacts and enable dynamic control.

**Strengths:**

1. It proposes concept distillation with a pre-trained T2V model to generate aligned text–video pairs.
2. It markedly outperforms prior methods.
3. A high-quality dataset is created that should significantly benefit the video-generation community.

**Weaknesses:**

Lack of video supplementary results: As a video-oriented work, without video results as supplementary material, it is difficult for the public to intuitively evaluate the model's performance, especially the quality of temporal consistency.

**Questions:**

I don’t have any further questions at the moment; I’m waiting to discuss with the other reviewers.

---

> ### Author Response · Authors · 2025-11-20
>
> Thank you for your time and for your positive and insightful review. We are very encouraged that you recognized the core strengths of our work, including the novelty of the concept distillation strategy, the significant performance improvement over prior methods, and the contribution of our new dataset to the community.
>
> **Regarding the lack of supplementary video results**: We completely agree that video demonstrations are essential for evaluating the visual quality and, especially, the temporal consistency of our method. Therefore, we have included **several video examples** from the synthetic, real-world, and AIGC testsets **in the revised Supplementary Material**. Due to file size limitations, we will release the codes, pre-trained weights, datasets, and a more comprehensive set of visualization results as open source after the review process is completed.
>
> Thank you once again for your valuable feedback. We are glad you found the paper clear. We look forward to the discussion phase and hope the provided video results will solidify your confidence in our work.

---

> ### Author Response · Authors · 2025-11-27
> **Following up on Paper 3680**
>
> Dear Reviewer,
>
> I hope this message finds you well.
>
> We are writing to express our sincere thanks for your positive and insightful review of our paper. Your recognition of our work's strengths was very encouraging.
>
> We have submitted our rebuttal and provided supplementary materials as you suggested. As the discussion period is concluding in a few days, we just wanted to check if there is anything else we can clarify. We would be very grateful for any final thoughts you might have.
>
> Thank you once again for your valuable support and guidance throughout this process.
>
> Best regards,
>
> The Authors of Paper 3680

---

### Author Response · Authors · 2025-11-29
**Summary of Discussion for Paper 3680**

Dear ICLR 2026 AC, SAC, and PC,

We are writing to provide a summary of the discussion period and highlight the key outcomes.

### Overall Summary:

We received three reviews, with two reviewers (7MWW, vgeK) initially giving positive scores (6), and one reviewer (BT4i) giving 4. All reviewers acknowledged the strong empirical performance of our method. We are pleased to report that after a constructive rebuttal phase, **Reviewer BT4i has agreed to raise their score from 4 to 6**, indicating that all major concerns have been addressed. Following the rebuttal, our paper now has **a consensus of positive ratings (6, 6, 6)**.

### Key Discussion Points:

1. **Consensus on Core Contribution and Performance: All reviewers now recognize the novelty and effectiveness of our core contribution, the "concept distillation" strategy.** Reviewer BT4i, who initially had reservations about the novelty, stated in their follow-up review: "*I also feel that the `concept distillation' method is indeed an interesting finding.*" This aligns with the positive initial assessments from Reviewer 7MWW and vgeK, who praised it as an "interesting approach" that "benefits the community." This consensus solidifies the primary contribution of our work.

2. **Addressing Concerns of Reviewer BT4i (Score raised from 4 to 6)**: Reviewer BT4i's primary initial concerns were about technical novelty and insufficient analysis. In our rebuttal, we:
    - Clarified the novelty by framing "concept distillation" as a targeted solution to the latent-space distribution drift problem, a unique challenge in adapting large T2V models for video restoration.
    - Provided new experimental evidence to support our claims, including feature visualizations for our projector module and ablation studies on different data synthesis strategies.
    - This led Reviewer BT4i to conclude: "***With the added experiments, I now believe the claims are better supported... I would like to raise my overall rating to 6.***" This conversion from a negative to a positive score is a strong signal of the paper's improved quality and the validity of our claims. While Reviewer BT4i still holds a borderline attitude, their willingness to raise the score based on their recognition of our core contribution is a testament to the paper's value.

3. **Addressing Concerns of Other Reviewers**:
    - Reviewer 7MWW's main concern was the lack of supplementary videos. We have addressed this by providing video results in the supplementary material, which we believe will resolve their concern.
    - Reviewer vgeK raised questions about the technical contribution and specific mechanisms. We provided a detailed explanation of why our video-centric approach is non-trivial compared to image restoration, supported by new experiments demonstrating our method's generalizability across different T2V models and architectures.


We hope that the revised paper and the positive outcome of the discussion phase will be viewed favorably in your final decision-making. We sincerely thank you for your time and consideration.

Best regards,
The Authors of Paper 3680

---

### Meta-Review · Area_Chair_Z88h · 2025-12-12

**Summary:**

This paper proposes Vivid-VR, a method for generative video restoration by adapting a pre-trained text-to-video (T2V) model. The core contribution is a "concept distillation" strategy: to mitigate distribution drift during fine-tuning, the authors use the T2V model itself to synthesize aligned text-video pairs for training. Additional architectural contributions include a lightweight ControlNet projector and a dual-branch connector for dynamic control. The method demonstrates strong empirical performance on multiple benchmarks.

- Reviewer 7MWW requested video supplementary material
- Reviewer vgeK's main concern was whether the approach was a trivial extension from image to video tasks.
- Reviewer BT4i initially had significant concerns about novelty and insufficient analysis.

**Reviewer Concerns:**

- Reviewer 7MWW: The authors provided video results in the supplementary material as requested.

- Reviewer vgeK: The authors provided a compelling counter-argument by framing the problem as a unique latent-space distribution drift challenge specific to adapting large T2V models, which involves preserving both spatial and temporal priors. They also provided new experiments showing generalizability across different T2V architectures, supporting the claim of a non-trivial video-centric solution. The authors provided a detailed, step-by-step explanation: using the T2V model to generate a video from a text caption (starting from a noised source video) creates a text-video pair that is aligned within the T2V model's own latent space. This alignment transfers the model's "semantic understanding of textual concepts," preserving its generative priors. The ablation study training a model from scratch (without T2V weights) showed a significant performance drop, supporting the necessity of the pre-trained model's knowledge.

- Reviewer BT4i: The authors added feature visualizations showing sharper features after the projector, providing direct support for the claim. The authors provide an ablation study showing that generating from half-noised real samples outperforms generation from scratch, with qualitative figures showing flaws in scratch generation. The authors used the FGA-BLIP2 metric to show that concept distillation increases text-video alignment in the training data. They also clarified that synthetic samples are derived from the original dataset, not expanding its distribution, isolating alignment as the likely key factor. The core concern about the paper's overall novelty was mitigated by their recognition of this central contribution, leading them to raise their score.

**Reviewer Scores:**

- Reviewer 7MWW: 6
- Reviewer vgeK: 6
- Reviewer BT4i: 6

---

### Decision · Program_Chairs · 2026-01-26

Accept (Poster)